# Parasites of *Sardinella maderensis* (Lowe, 1838) (Actinopterygii: Clupeidae) and Their Potential as Biological Tags for Stock Identification along the Coast of West Africa

**DOI:** 10.3390/biology12030389

**Published:** 2023-02-28

**Authors:** Abdou Matinou Ogbon, Richmond Afoakwah, Kwadwo Kesse Mireku, Nounagnon Darius Tossavi, Ken MacKenzie

**Affiliations:** 1Centre for Coastal Management-Africa Centre of Excellence in Coastal Resilience, University of Cape Coast, Cape Coast PMB TF0494, Ghana; 2Department of Fisheries and Aquatic Sciences, University of Cape Coast, Cape Coast PMB TF0494, Ghana; 3Laboratoire de Parasitologie et Ecologie Parasitaire, Université d’Abomey-Calavi, Cotonou 01 BP 526, Benin; 4Department of Forensic Sciences, University of Cape Coast, Cape Coast PMB TF0494, Ghana; 5Ecole d’Aquaculture, Université Nationale d’Agriculture, Porto-Novo 01 BP 55, Benin; 6School of Biological Sciences (Zoology), University of Aberdeen, Tillydrone Avenue, Aberdeen AB24 2TZ, Scotland, UK

**Keywords:** parasite fauna, biological tags, stock identification, *Sardinella maderensis*, Benin, Ghana, West Africa

## Abstract

**Simple Summary:**

*Sardinella maderensis*, representing one of the most commercially important small pelagic fish species along the coast of West Africa, is suffering a drastic decline as a result of overfishing, the overcapacity of fishing fleets, the destruction of fish habitat, the use of inappropriate fishing gears and techniques, as well as environmental changes. The limited reliable information on their stock for sustainable management constitutes one of the problems facing small pelagic fish in general and *Sardinella maderensis* particularly along the coast of West Africa. The key goals of this study were to identify the parasites of *Sardinella maderensis* and to assess their potential use as biological tags for stock identification along the coast of West Africa (Benin and Ghana). The objectives of this study were to determine the morphological parameters (total length and body weight) of *S. maderensis*, identify their parasites in Benin and Ghana and select appropriate parasites with potential to be used as biological tags. The results suggest that the nematode *Anisakis* sp(p). and the cestode *Tentacularia coryphaenae* may serve as potential biological tags for the stock identification of *Sardinella maderensis.*

**Abstract:**

This study is the first to provide information on the parasite fauna of *Sardinella maderensis* along the coasts of Benin and Ghana, and the first to investigate the potential use of parasites as biological tags in fish population studies in the area. It may thus serve as a starting point for upcoming studies. From February to June 2021, a total of 200 *S. maderensis* were sampled from the fishing port of Cotonou (Benin) and the Elmina landing site (Ghana). The prevalence and abundance of each parasite were recorded. The following are the outcomes of this study: Parasite species, such as *Parahemiurus merus*, *Mazocraeoides* sp. and *Hysterothylacium fortalezae*, were recorded along the coasts of Benin and Ghana, while *Anisakis* sp(p). and *Tentacularia coryphaenae* were only recorded along the coast of Benin. *Parahemiurus merus* was the most prevalent and abundant among all the parasites recorded. *Anisakis* sp(p). and *T. coryphaenae* were selected as having potential in the stock identification of *S. maderensis*. Both parasites were only recorded along the coast of Benin at a low prevalence. As a result, examinations of more *S. maderensis* from each location for these parasites may justify their use in stock identification studies.

## 1. Introduction

It has been recognized that the study of parasites in fish in sub-Saharan Africa needs greater attention, especially given the considerable aquaculture and wild-caught fishery industries found across the continent. Studies on marine fish parasitology have so far mainly been focused on parasite population surveys and new species identification. Research on the impact of parasites related to economically harvested fish in this region, or on how parasite data can be used to enhance fisheries management, is limited [1].

*Sardinella maderensis* and *S. aurita* represent the most abundant and commercially important species of small marine pelagic fish along the coast of West Africa [2]. Together, they account for more than 40% and 16.2% of total landings in Ghana and Benin, respectively, with *S. maderensis* being the most abundant fish species in Benin [3,4]. The lack of reliable data on stock structure for the management of small pelagics in general, and these two *Sardinella* sp(p). in particular, is a significant problem in this region. This problem can be addressed by providing fishery managers and scientists with reliable data using multiple methods of stock identification. The present study thus seeks to provide an alternative low-cost method for the stock identification of these marine pelagic species through the use of parasites as biological tags to augment existing methods.

The basic principle underlying the use of parasites as tags in fish population studies is that fish can become infected with a parasite only when they are within the endemic area of that parasite. The endemic area is the geographic region in which conditions are suitable for the transmission of the parasite, including biotic factors, such as the presence of other hosts essential for the completion of the parasite’s life cycle, and abiotic factors, such as temperature and salinity. If infected fish are found outside the endemic area of the parasite, we can infer that these fish had been in the parasite’s endemic area at some time in the past [5]. Fish can thus be said to carry a “parasitological fingerprint” by which their past movements can be traced. Various authors have listed criteria or guidelines for the selection of parasites suitable for use as biological tags in fish population studies [5]. The most important of these is that the parasite should have a lifespan in the target host appropriate to the nature of the study. For stock identification studies, this means a lifespan of more than one year. The fish parasites that best meet this criterion are the larval stages of helminths, such as trematode metacercariae and larval nematodes and cestodes. These are the “resting” stages in the fish host, which may exist in this state for many years. The efficiency of the parasite tag approach thus relies on sufficient information on the biology and ecology of the parasite, particularly with regard to its life cycle and its lifespan in the fish. A lack of such information was earlier recognized as a limiting factor, but with the increase in studies of marine parasite biology in recent years, the resulting information has greatly increased the efficiency of the method. The use of parasites as biological tags in fish population studies has now become a widely accepted method of stock identification [6].

The use of parasites as biological tags has the following advantages over other methods of stock identification:It is more appropriate for studies of small delicate species of fish, such as small clupeoids, for which artificial tags can be used with difficulty, or not at all.Using parasite as tags is more cost-effective than artificial tagging because fish samples can be obtained from the routine sampling of commercial or research vessel catches without the need for costly dedicated sampling programs.The use of parasite tags has an advantage over the use of host genetics because it can often be used to identify subpopulations of fish distinguished by behavioral differences, but between which there is still a considerable amount of gene flow which can render genetic studies inconclusive.

A recent stock identification study of South African sardines *Sardinops sagax* is an excellent example of what can be achieved in the fishery management of a small pelagic species using parasites as biological tags [7,8]. Using parasites in this way proved to be a powerful tool for population structure studies of these sardines and provided more convincing support for a multiple-stock hypothesis than other methods of stock identification [8]. Using this study as an example, the aim of the present study is to present the results of a preliminary survey carried out to determine what parasites infect the target host in the study area. We also identify those parasites of *S. maderensis* with the potential to be used as biological tags for stock identification, the null hypothesis being that all fish populations in the study area belong to a single stock.

## 2. Materials and Methods

### 2.1. Sample Collection

A total of 200 specimens of *S. maderensis*, consisting of 100 specimens each from Cotonou fishing port in Benin (6° 21′4.212″ N, 2°25′58.296″ E) and the Elmina landing site, Elmina, Ghana (5°04′57.3″ N 1°21′02.6″ W) (Figure 1), were obtained from artisanal catches in 2021 from February to June. The specimens were kept on ice and transported to the laboratory. The Benin samples were analyzed at the laboratory of Parasitology and Ecology of Parasites of the Department of Zoology, the University of Abomey-Calavi, and the Ghana samples were analyzed at the laboratory of the Department of Fisheries and Aquatic Sciences, University of Cape Coast. These two study areas were chosen because of the existence of two nurseries of *S. maderensis* along the coast of Ghana. The first nursery is located in the East coast of Ghana, a shared stock by Togo and Benin and the second nursery in the west coast of Ghana, a shared stock with la Côte d’Ivoire [9].

### 2.2. Morphological Data

The total length (TL) of the fish was measured as the length from the snout to the most posterior part of the caudal fin. The total lengths were measured to the nearest 0.1 cm using a measuring board. The body weight (BW) of the fish were measured to the nearest 0.1 g by placing the fish on an ADAM scale electronic balance. The samples were sexed by opening and observing the characteristics of the gonads.

### 2.3. Parasite Collection

The protocol for parasite collection used in this study was that of the book *Parasites of Marine Fish and Cephalopods* [10]. Ectoparasites were examined macroscopically on the fish’s body surface and apertures (eyes, skin, fins, gills, nostrils, anus, mouth cavity and fins) using a hand lens and an AmScope dissecting microscope at 30X magnification. The mucus was scraped from the skin, fins, nasal pits, gills and the internal portion of the operculum, and was examined for ectoparasites under a Motic microscope at 10X and 40X magnification. The eyeballs were removed and then punctured with a syringe to extract the eye fluid. The eye fluid was examined under the Motic microscope for digenean metacercaria.

For endoparasites, the fish specimens were dissected by applying four incisions. The first incision was made vertically from the anus to end of the lateral line, whereas the second incision was made through the end of the lateral line to the beginning of the upper operculum bone. The third incision was made from the beginning of the upper operculum bone to the lower operculum bone, and the final incision made horizontally through the ventral portion of the fish to the lower of the operculum bone. The dissected parts of the fish were removed, and the organs were exposed. The viscera were split into the stomach, pyloric caeca, intestine, gonad, gall bladder, liver, kidney and spleen. All the organs were removed, placed in labelled Petri dishes and filled with a 0.9% saline solution. The stomach, intestine and pyloric caeca were opened longitudinally, and the contents were scraped and examined for parasites. The gall bladder was punctured, and the bile was examined for parasites. A smear of the liver, pylorus, kidney and spleen were prepared by cutting a small piece of each organ before being gently pressed with the back of the forceps on a microscope slide and examined under the Motic microscope at 40X magnification.

### 2.4. Parasite Preparation and Preservation

All the parasites recorded in this study, except nematodes, were counted and fixed in 70% ethanol. They were stained with borax carmine and cleared with eugenol (clove oil), whereas the nematodes were cleared with glycerin. All the parasites were mounted in Canada balsam and viewed under the Motic microscope when dry at different magnifications based on the size of specimen. All the images captured were used for taxonomical identification.

### 2.5. Data Analysis

All the data collected in this study were skewed, hence non-parametric tests were performed.

#### 2.5.1. Morphological Data Analysis

For morphometrics data, a Mann–Whitney U test was performed to determine whether the fish total length and weight were significantly different across all the sampling locations. Additionally, a Kruskal–Wallis test was conducted to determine if the fish total lengths significantly differed among sexes (male, female and indeterminate), followed by a pairwise post-hoc Dunn’s test for multiple comparison with Bonferroni adjustments.

#### 2.5.2. Parasitological Data Analysis

For parasitological data, prevalence P(%) and mean abundance (MA) of infection were calculated according to [11].
$$P\left(\%\right)=\frac{ni}{n\Sigma }\times 100$$
where *ni* = number of hosts with a specific parasite *i* and *n*∑ = total number of hosts examined.
$$MA=\frac{I\Sigma_{i}}{n\Sigma }$$
where *I*∑*_i_* = total number of a specific parasite species *i* and *n*∑*_i_* = total number of examined hosts [10].

The prevalence of parasites were compared among locations using unconditional exact tests [12]. Abundances were compared between localities using the bootstrap *t*-test. The biased accelerated bootstrap (BCa bootstrap) was used to provide the confidence interval of the mean abundance. The length classes of the fish were compared with parasite abundance using the Mann–Whitney U test. The comparison between the sex categories (males, females and indeterminate) and the abundance of parasites was performed using the Kruskal–Wallis test. Spearman’s correlation was conducted to find the relationship between the abundance of parasites and the length classes. All statistical analyses were performed using the Quantitative Parasitology Web portal (https://www2.univet.hu/qpweb/qp10/ (accessed on 28 July 2022)) [12] and the Statistical Package for Social Science (SPSS) 2019 version 26. The significance level was set at *p* < 0.05.

## 3. Results

### 3.1. Morphological Data

Specimens from Benin varied from 14.5 to 32.2 cm in total length with a mean length of 17.70 ± 2.97, whereas those from Ghana varied from 16.00 to 32.00 cm in total length with a mean length of 16.00 ± 3.98. The body weight of the specimens from Benin varied from 28.00 to 287.00 g with a mean body weight of 144.95 ± 47.25, and those from Ghana varied from 38.46 to 258.92 g with a mean body weight of 120.53 ± 57.41 (Table 1).

In Benin, 50% of the sampled specimens were males, while females and indeterminates constituted 47% and 3%, respectively. In Ghana, the males represented 41% of the sampled specimens, while the indeterminates and females represented 38% and 21%, respectively (Table 1).

#### 3.1.1. Comparison between Fish Total Length and Body Weight across Sample Locations (Benin and Ghana)

A Mann–Whitney U test demonstrated that the fish recorded in Benin were significantly longer (Median = *Mdn* = 25.10 cm, *n* = 100) than those recorded from Ghana (*Mdn* = 23.00 cm, *n* = 100), (*p* = 0.001), with a small effect size r = 0.29. Additionally, this same test showed that the fish recorded in Benin were significantly heavier (*Mdn* = 140.00 g, *n* = 100) than those recorded in Ghana (*Mdn* = 119.14 g, *n* = 100), (*p* = 0.003), with a small effect size r = 0.21.

#### 3.1.2. Comparison between the Fish Total Lengths and Body Weight across Sex Categories (Male, Female and Indeterminate Sex)

A Kruskal–Wallis test showed a significant difference in fish total lengths across sexes (*p* = 0.001). This same test demonstrated a significant difference in fish body weight across sexes (*p* = 0.001). A pairwise post-hoc Dunn test with Bonferroni adjustments indicated that Indeterminate sex were observed to be significantly different from males (x^2^
*=* 80.08; *p* = 0.001) and females (x^2^
*=* 110.10; *p* = 0.001) in terms of fish total lengths. Additionally, there was a significant difference between males and females (x^2^
*=* 30.08; *p* = 0.004) in terms of fish length. Therefore, all the sexes differed significantly from each other in terms of fish total lengths (Figure 2). Furthermore, this same test showed that the indeterminate sex were observed to be significantly different from males (x^2^
*=* 79.57; *p* = 0.001) and females (x^2^
*=* 110.66; *p* = 0.001). Additionally, there was a significant difference between males and females (x^2^
*=* 31.09; *p* = 0.002). Therefore, all the sexes differed significantly from each other in terms of fish body weight (Figure 3).

### 3.2. Parasite Data

A total number of 466 parasite specimens, consisting of 313 digeneans (*Parahemiurus merus*), 68 monogeneans (*Mazocraeoides* sp.), 78 nematodes (64 *Hysterothylacium fortalezae* and 14 *Anisakis* sp(p).), and 7 cestodes (*Tentacularia coryphaenae*), were recorded during this study. The digenean *Parahemiurus merus* was the most prevalent among all the parasites found in this study, with a frequency of occurrence of 45% in Benin and 21% in Ghana, with corresponding mean abundances of 1.63 ± 2.76 and 0.75 ± 2.08 in Benin and Ghana, respectively (Table 2).

#### 3.2.1. Comparison of Parasite Prevalence and Mean Abundance of Infection across Sampling Locations

The prevalence and mean abundance of *P. merus* recorded in Benin were higher than those recorded in Ghana (Table 2). The unconditional exact test revealed that the prevalence of *P. merus* recorded in Benin differed significantly from that recorded in Ghana (*p* < 0.05). A bootstrap two-sample t-test based on 2000 bootstrap replications showed that the mean abundance of *P. merus* in Benin differed significantly from that in Ghana (*p* < 0.05) (Table 3).

The prevalence and mean abundance of *H. fortalezae* recorded in Ghana were all higher than those recorded in Benin (Table 2). However, the unconditional exact test revealed that there was no significant difference in the prevalence of *H. fortalezae* in Benin and that in Ghana (*p* > 0.05) (Table 3). The mean abundances of *H. fortalezae* were not compared due to low numbers.

The prevalence and mean abundance of *Mazocraeoides* sp. recorded in Ghana were higher than those recorded in Benin (Table 2). The unconditional exact test showed that the prevalence of *Mazocraeoides* sp. in Ghana differed significantly from that in Benin (*p* < 0.05). A bootstrap two-sample t-test based on 2000 bootstrap replications revealed that there was no significant difference between the mean abundances of *Mazocraeoides* sp. in Ghana and Benin (*p* > 0.05) (Table 3).

#### 3.2.2. Comparison of Parasite Prevalence and Mean Abundance of Infection across Fish Length Classes

Two length classes (≤25 and <25 cm) were selected from the total length data (14.5–25.0 cm) and (25.01–32.2 cm). *Parahemiurus merus* and *T. coryphaenae* were more prevalent in the length class > 25.00 cm than in the length class ≤25.00 cm. However, *H. fortalezae*, *Anisakis* sp(p). and *Mazocraeoides* sp. had higher a prevalence in fish of length class ≤25.00 cm (Figure 4). Further analysis indicated that the abundance of *P. merus* was significantly different across the two length classes (*p* < 0.05) with a small effect size r = 0.21.

#### 3.2.3. Comparison of Parasite Prevalence and Mean Abundance of Infection across Fish Sex Categories

Among the sexes, the males of *S. maderensis* had the highest prevalence values for *P. merus* and *Anisakis* sp(p). compared to the female and indeterminate sex. Conversely, females had the highest prevalence for *T. coryphaenae* compared to the male and indeterminate sex. *Hysterothylacium fortalezae* was most prevalent in the indeterminate sex category. On the other hand, the female and indeterminate sex of *S. maderensis* had a higher prevalence of *Mazocraeoides* sp. than males (Figure 5). The Kruskal–Wallis test showed that the abundance of *P. merus* was significantly different between the sexes (Kruskal–Wallis test: N = 200; df = 2; x^2^ = 24.01; *p* < 0.05). The Kruskal–Wallis test showed that the abundance of *P. merus* was significantly different between the sexes (Kruskal–Wallis test: N = 200; df = 2; x^2^ = 24.01; *p* < 0.05). A pairwise post-hoc Dunn test with Bonferroni adjustments indicated that the indeterminate sex was significantly different from males (x^2^
*=* 31.83; *p* < 0.05) and females (x^2^
*=* 44.62; *p* < 0.05) in terms of the abundance of *P. merus*. However, there was no significant difference between the males and females (x^2^
*=* 12.79; *p* > 0.05). Therefore, the indeterminate sex category differed significantly from females and males in terms of the abundance of *P. merus*, whilst male and female were not significantly different (Figure 6).

#### 3.2.4. Relationship between Abundance of Parasites and Fish Length

Spearman’s correlation test showed a significant but weak positive linear relationship between the fish lengths and the abundance of *P. merus* only (r = 0.21, *p* < 0.05) (Figure 7).

## 4. Discussion

### 4.1. Fish Morphological Data

The results obtained from this study show that the total length of *S. maderensis* collected along the coast of Ghana ranged from 16.00 to 32.00 cm, while those collected along the coast of Benin ranged from 14.50 to 32.20 cm. These total lengths were similar to those obtained for *S. maderensis* (14 to 32 cm) along the coast of Benin [4] and longer than those recorded for the same fish species (9.8 to 28.2 cm TL) along the coast of Ghana [13]. They were, however, smaller than those recorded from the Liberian coast (5.5 to 42 cm TL) [14].

The body weights of the fish recorded in this study were greater than those recorded along the Nigerian coast (9.73 to 39.55 g) [15] and the south-west of Turkey (10.8 to 73 g) [16]. These variations in total length and body weight recorded in this study may be due to environmental factors, such as temperature, salinity, and food availability, as well as genetic diversity.

### 4.2. Parasitological Data

In the present study, four parasite groups (Monogenea, Digenea, Cestoda and Nematoda) and five genera of parasites (*Parahemiurus merus*, *Hysterothylacium fortalezae*, *Anisakis* sp(p)., *Tentacularia coryphaenae* and *Mazocraeoides* sp.) were recorded along the coasts of Benin and Ghana. The digenetic trematode *P. merus* was the most predominant among the parasite species infecting *S. maderensis* in the two sampling areas. This parasite has been recorded in many marine fish species worldwide, including clupeids [17]. It was reported previously in *Sardinella cameronensis* (*S. maderensis*) along the coasts of Ghana [18] and Senegal [19], from *Sardinella aurita* from the Gulf of Gabès, Tunisia [20,21] and from the Algerian coast [22].

The prevalence of *P. merus* in Benin (45%) was higher than that in Ghana (21%) during the study. The prevalence of this species in Benin and Ghana was lower than that recorded from *S. aurita* in Bizerte (84%), Kelibia (84.44%), Mahdia (48.05%) and Zarzis (86.84%) off the coast of Tunisia [21], but higher than that recorded from *S. aurita* from Gabès (11.57%) off the coast of Tunisia [21] and the Algerian coast (5.31%) [22]. The high prevalence of *P. merus* recorded in this study may be due to the abundance of its intermediate hosts in the study area. The life cycle of *P. merus* remains unknown, but metacercariae have been reported from chaetognaths [23]. Gastropod molluscs and copepods are presumed to be the primary and secondary intermediate hosts of *P. merus* [24]. However, known invertebrate hosts were not examined in this study.

Two nematode parasites were recorded in *S. maderensis*, namely the third larval stages of *Hysterothylacium fortalezae* along the coasts of Benin and Ghana and *Anisakis* sp(p). in Benin. *Hysterothylacium fortalezae* larvae were previously reported from some midwater and benthopelagic stomiiform fish in the northern Gulf of Mexico [25], and from *Selene setapinnis* in the state of Rio de Janeiro, Brazil [26]. It was also recorded from *Percophis brasiliensis* in the municipality of Niterói, Rio de Janeiro, Brazil [27]. However, it has not been reported from any fish along the coast of West Africa. The only known definitive host of *H. fortalezae* is the serra Spanish mackerel *Scomberomorus brasiliensis* [28]. However, the West African Spanish mackerel *Scomberomorus tritor* may be a likely host in the present study area.

The prevalence of *H. fortalezae* recorded in this study in fish collected along the coast of Benin (2%) and Ghana (4%) were lower compared to those recorded from *Selene setapinnis* (26.7%) in the northern Gulf of Mexico [26] and from *Percophis brasiliensis* (21.87%) in the municipality of Niterói, Rio de Janeiro, Brazil [27]. However, the prevalence of *H. fortalezae* in specimens from Ghana (4%) was higher compared to those recorded from *Pollichthys mauli* and *Polyipnus clarus* (3% and 1%, respectively) in the northern Gulf of Mexico [25].

The third-stage larvae of *Anisakis* sp(p). were found only in Benin at a low prevalence of 5%. They have also been reported in *S. maderensis* off the coast of Nigeria with a prevalence of 2% [29]. The genus *Anisakis* currently comprises nine “cryptic” or “sibling” species [30], which are very similar in morphology and usually rely on molecular methods for specific identification. Cetaceans, mainly toothed whales, are the definitive hosts of *Anisakis* sp(p). Pelagic crustaceans are the first intermediate hosts, while larger crustaceans, particularly euphausiids, and smaller fish species are thought to be the important second intermediate hosts. Larger fish and cephalopods serve as paratenic hosts [31,32].

A monogenean parasite of the genus *Mazocraeoides* was recorded from the gills of *S. maderensis* in both study areas. This monogenean genus is relatively diverse in species and infects many species of clupeid fish [33,34]. A species of *Mazocraeoides* was found infecting *Sardinella longiceps* from the Visakhapatnam coast, Bay of Bengal in India [34], but none appear to have been previously reported from *S. maderensis,* so this may be a new species. The genus is characterized by a broad body and clamps arranged along the lateral margins of the body with the anterior pair anterior to the level of the ovary [35].

The plerocercoid of the trypanorhynch cestode *Tentacularia coryphaenae* was found only along the coast of Benin. The Trypanorhyncha is the most species-rich order of cestodes infecting elasmobranch fish as definitive hosts. Larval trypanorhynchs, of as many as 14 genera, have been reported from second intermediate hosts, mostly teleosts, but the identities of the elasmobranch hosts of many of these larvae have yet to be established. Various invertebrate groups, as well as possibly fish, apparently serve as the first intermediate hosts [36]. While most trypanorhynch species are fairly host-specific, at least as adults, *T. coryphaenae* has been reported from 11 different shark species and its larvae has been reported from more than 60 teleost species [37], but this appears to be the first record from *S. maderensis.*

The prevalence of *T. coryphaenae* (6%) recorded along the coast of Benin was lower than that recorded off the coast of South Africa in oilfish (*Ruvettus pretiosus*) (100%) and snoek (*Thryrsites atun*) (49.7%) [38,39], and in the black scabbardfish (*Aphanopus carbo*) (25.8%) from Portuguese waters [40]. However, the prevalence was higher than those recorded from *Scomber japonicus* (2%) and South African sardines (*Sardinops sagax*) (1%) off the coast of South Africa [7,41]. The absence of *T. coryphaenae* in the fish sampled from Ghana may be a result of the low number of fish sampled. It may also be related to the differential occurrence of suitable shark definitive hosts in the two study areas.

#### 4.2.1. Relationship between Abundance of Parasite and the Fish Sizes

In this study, a weak positive relationship was found between the abundance of *P. merus* and the length of *S. maderensis*, which implies that the abundance of *P. merus* increases with fish length. A positive relationship was also reported from *Anchoa tricolor*, *Spagrus spagrus* and *Opisthonema oglinum* along the coast of Brazil [42,43,44]. This may be due to the fact that larger *S. maderensis* consume greater numbers of the intermediate prey organisms of this parasite, [45,46]. Both juvenile and adult *S. maderensis* show a preference for crustaceans [47,48].

#### 4.2.2. Parasites Selected as Potential Biological Tags

Two parasite taxa, *Anisakis* sp(p). and *T. coryphaenae*, were selected following established guidelines [5] as potentially useful biological tags for the stock identification of *S. maderensis* along the coast of West Africa. Both parasites have long-lived larval stages that survive as “resting” stages in their fish intermediate hosts for several years, possibly for as long as the infected host lives. *Anisakis* sp(p). larvae have proved to be amongst the best tag parasites for small pelagic fish [49], and *T. coryphaenae* was selected by [50] as potentially the most valuable biological tag for the stock identification of skipjack tuna, *Katsuwonus pelamis*. As the genus *Anisakis* comprises a group of nine species, each with its own cetacean host preferences, it is crucially important to identify the species using molecular methods. This identification can then be related to the known occurrence of the cetacean host(s) in the study area. This approach cannot be used for *T. coryphaenae* because of its wide host specificity to both fish intermediate and definitive hosts, but statistically significant variations in the prevalence and abundance between sampling areas may be indicative of different dietary compositions between fish populations [51]. However, the hosts’ stomach content analyses were not examined in this study. For example, the occurrence of *T. coryphaenae* in Benin but not in Ghana may be related to differences in the diet of *S. maderensis* between the two areas. *Parahemiurus merus* and *Mazocraeoides* sp. have short-lived adult stages and are therefore not considered useful as biological tags for fish stock identification, but they may be useful in seasonal migration studies. More information is needed on the life cycle of *H. fortalezae*, particularly regarding the identity of its definitive host(s) in the study area, before its potential as a biological tag can be assessed.

## 5. Conclusions

The present study intended to apply parasite data to the stock identification of *Sardinella maderensis*, one of the most valuable small pelagic fish species along the coast of West Africa. *Tentacularia coryphaenae* and *Anisakis* sp(p). were found to have potential for the future stock identification of *S. maderensis* along the coasts of Benin and Ghana and adjacent areas in West Africa. Even though the prevalence of these two parasites were low, the fact that they have only been found in fish sampled from one of the two localities is promising. Examinations of more *S. maderensis* for *Anisakis* and *T. coryphaenae* from Ghana and Benin, and from adjacent areas, would clarify the distribution of each parasite. It is also important to take samples at different seasons to check for seasonal migratory patterns. The genetic identification of the species of *Anisakis* present in the samples is essential as its distribution could then be related to the known distribution of the cetacean hosts. Finally, due to the limited research on marine parasitology in West Africa, there is an urgent need to continue exploring this area of study.

## Figures and Tables

**Figure 1 biology-12-00389-f001:**
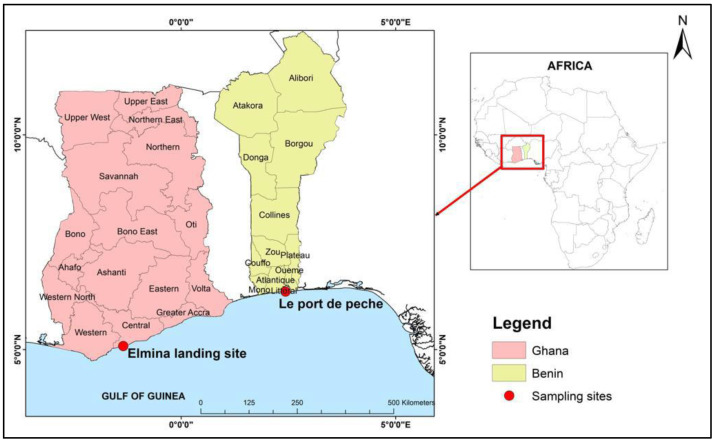
Map of the study area showing Elmina landing site in Ghana and le Port de pêche de Cotonou (fishing port of Cotonou) in Benin.

**Figure 2 biology-12-00389-f002:**
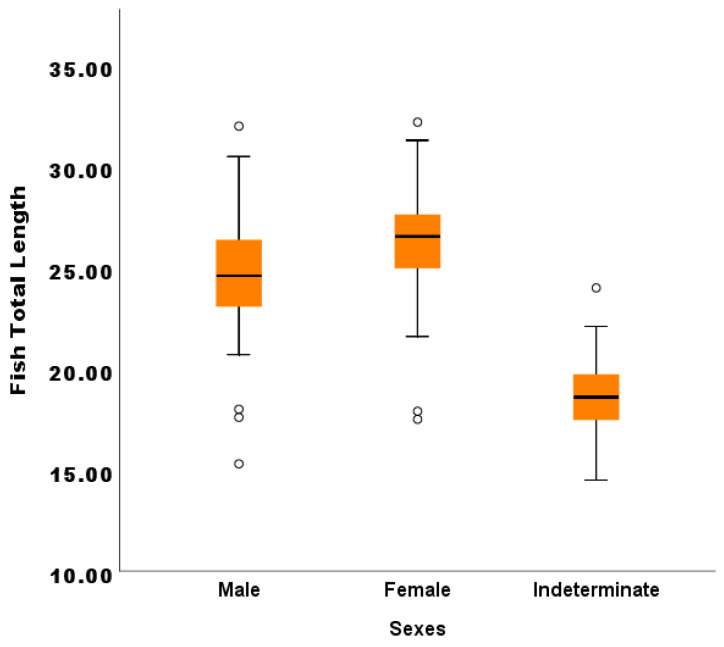
Overall total length of *S. maderensis* in different sex categories for all samples from the study areas.

**Figure 3 biology-12-00389-f003:**
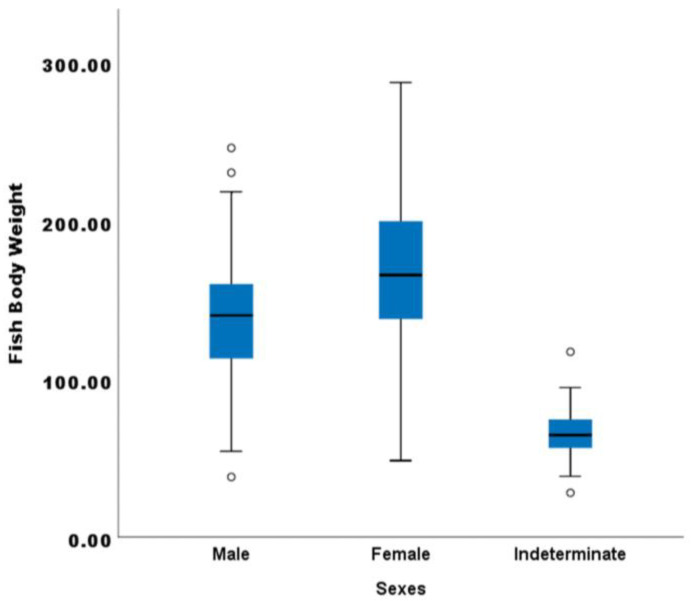
Overall body weight of *S. maderensis* in different sex categories for all samples from the study areas.

**Figure 4 biology-12-00389-f004:**
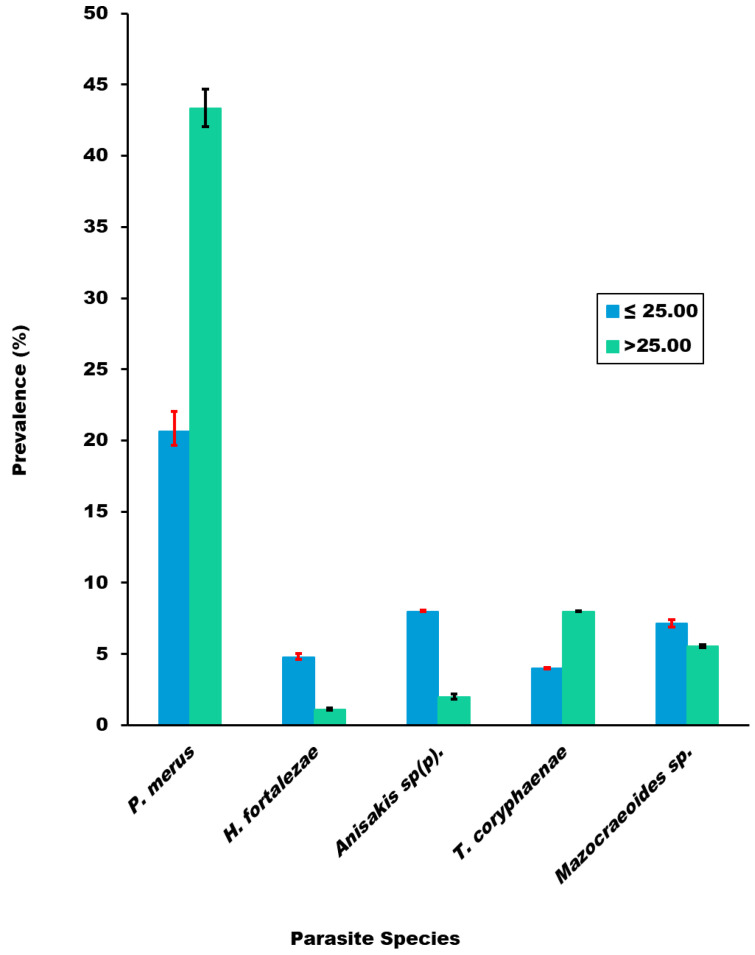
Overall prevalence of parasite taxa in *S. maderensis* of length ≤25 cm and those >25 cm collected from Benin and Ghana.

**Figure 5 biology-12-00389-f005:**
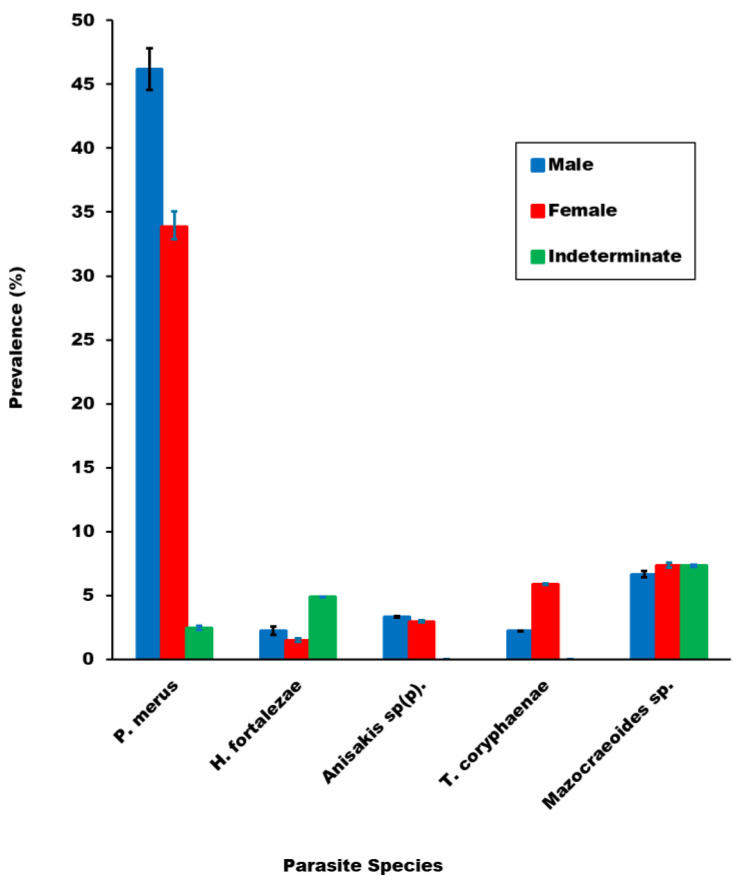
Overall parasite prevalence in *S. maderensis* in different sex categories for all samples from both study areas.

**Figure 6 biology-12-00389-f006:**
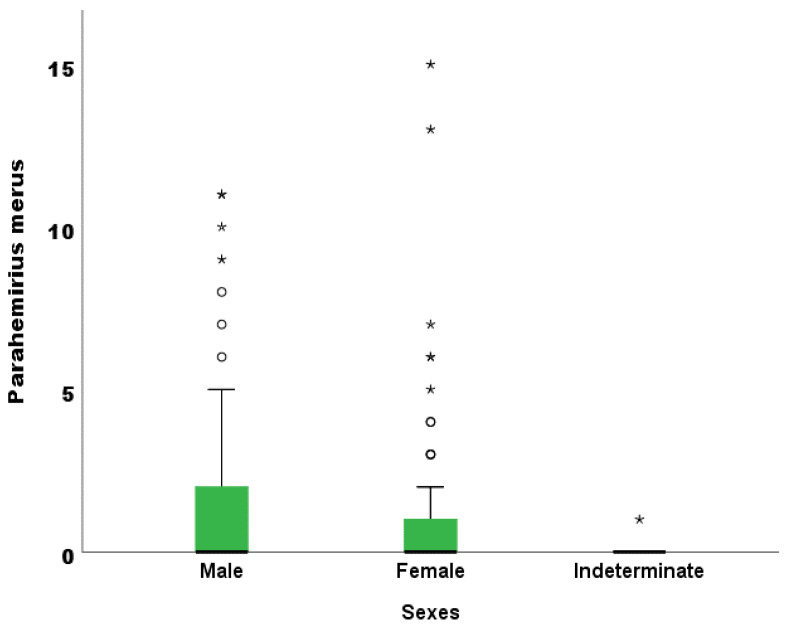
*Parahemiurus merus* abundance in *S. maderensis* in different sex categories for all samples from both study areas.

**Figure 7 biology-12-00389-f007:**
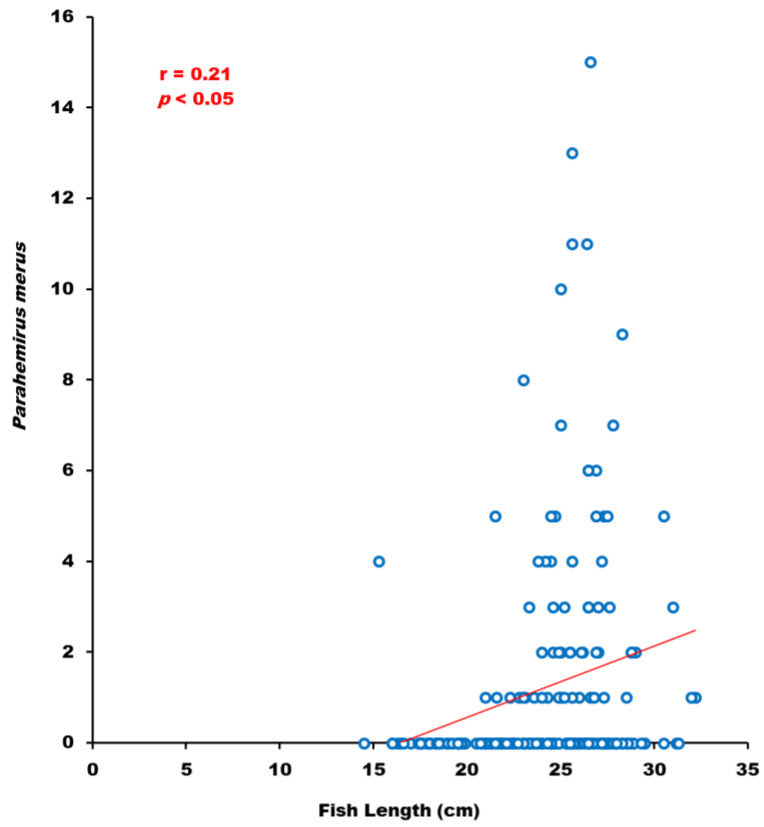
Relationship between abundance of *P. merus* and fish length.

**Table 1 biology-12-00389-t001:** Summary statistics of the fish morphometrics data: N, Number of fish; Sex (M, Male; F, Female; I, indeterminate); TL, Total Length; BW, Body Weight.

Sampling	N	Sex			TL Range (cm)	BW Range (g)
Locations		M	I	F	(Mean ± SD)	(Mean ± SD)
Benin	100	50	3	47	14.5–32.20	28.00–287.00
					(17.70 ± 2.97)	(144.95 ± 47.25)
Ghana	100	41	38	21	16.00–32.00	38.46–258.92
					(16.00 ± 3.98)	(120.53 ± 57.41)

**Table 2 biology-12-00389-t002:** Summary statistics for the parasites found recovered from *S. maderensis* along the coast of West Africa (Benin–Ghana): P (%), Prevalence; MA, Mean Abundance; n, Number of fish specimens sampled; SI, Site of infection; S, Stomach; L, Liver; V, Visceral; G, Gills.

Parasites	SI	Benin (Cotonou)		Ghana (Elmina)	
		(*n* = 100)		(*n* = 100)	
		P (%)	MA ± SD	P (%)	MA ± SD
			(95% CI)		(95% CI)
**Digenea**					
*P. merus*	S	45	1.63 ± 2.76	21	0.75 ± 2.08
			(1.19–2.3)		(0.43–1.25)
**Nematode**					
*H. fortalezae* *	S	1	0.06 ± 0.60	4	0.3 ± 2.28
			(0–0.18)		(0.03–1.3)
*Anisakis* sp(p).	S/L	5	0.14	0	0
			(0.03–0.34)		
**Cestode**					
*T. coryphaenae* *	V	6	0.07 (0.02–0.13)	0	0
**Monogenea**					
*Mazocraeoides* sp.*	G	3	0.08 ± 0.55	11	0.3 ± 1.19
			(0.01–0.27)		(0.13–0.67)

* New host records.

**Table 3 biology-12-00389-t003:** Parasites displaying significant differences in prevalence (%) and/or mean abundance in S. maderensis.

Parasites	Prevalence (%)			Mean Abundance		
	Benin	Ghana	*p*-Value	Benin	Ghana	*p*-Value
**Digenea**						
*P. merus*	45	21	<0.05 *	1.63	0.75	<0.05 *
**Nematoda**						
*H. fortalezae*	1	4	>0.05	0.06	0.3	-
**Monogenea**						
*Mazocraeoides* sp.	3	11	<0.05 *	0.08	0.3	>0.05

* Signifies *p*-values significantly difference.

## Data Availability

Data will be made available on demand.

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
