# Peer review of "Parasites of Sardinella maderensis (Lowe, 1838) (Actinopterygii: Clupeidae) and Their Potential as Biological Tags for Stock Identification along the Coast of West Africa"

_biology, 2023, doi:10.3390/biology12030389_

Round 1

Reviewer 1 Report

The authors present new data regarding parasitism in different fish populations on the coast of Africa. They correlate the existence of such parasites with the populations proposing novel tags for fish studies. 

The study is well-organized and quite clear. Nevertheless, some minor changes may be addressed. They could include as supplementary some figures showing pictures of fish from the 2 areas of study to prove that is a need for a tag, and that are not distinguished by the naked eye. Besides, material and methods could also be further explained for example, the part of the data analysis. Authors should, at least, include a sentence indicating that for each case the statistical test used will be described in the specific section. 

Author Response

Dear Reviewer,

Thank you for spending your valuable time reviewing our manuscript.

Kindly find attached the point-by-point response to your comments.

Kind regards,

Abdou.

Reviewer 2 Report

Line 2: The title of the manuscript should be amended by adding the Latin species name of the author in parentheses - (Lowe, 1838) and then adding to it the classification (if you wish) - (Actinopterygii: Clupeidae). Thus, the reviewer believes that the title of the manuscript could be: Parasites of Sardinella maderensis (Lowe, 1838) (Actinopterygii: Clupeidae) and their potential as biological tags for Stock Identification along the coast of West Africa. Otherwise, the Latin name is given incorrectly.

Line 27, 37: Anisakis sp(p). -> Anisakis spp. If we are talking about several species belonging to the same genus and without specifying specific species, we should write spp.

Line 30: It is necessary to write Sardinella maderensis in Abstract for the first time, and only then to resort to the abbreviated name.

Line 122: The names of regions in countries are very hard to read in Figure 1. This gives Figure 1 the impression of being overloaded with information. A new font formatting is needed for a better perception.

Line 133: "Cephalopods"[10]". - A text break is needed - Cephalopods" [10]

Line 189: "The significant level was set at P < 0.05" → p < 0.05. P is the accepted designation for probability. The significant level is recommended to denote by the letter p written in italics. And then everywhere else in the manuscript where it refers to the significance level, not the probability.

Line 202: Table 1 “144.95 ±47.25” → 144.95±47.25

Line 212: "3.1.1. Comparison between the fish total lengths and body weights across sex categories" Please note that item 3.1.1 has already been used on Line 204. Please rename the paragraph.

Line 274: "3.3. Parasite Data" Please note that item 3.2 is not in the text!

Line 312: p-value

Line 327: Anisakis spp.

Line 390: Benin [4]

Line 391: Ghana [13]

Line 395: “39.55g” → 39.55 g

Line 402, 420, 435: spp.

Line 462: [40]

Line 500: spp.

There is a need to format Figures 2-7. Inscriptions are poorly readable. It is not clear why the authors made some of the Figures black and white and some in color (Figure 5). It would be good to bring Figures 2-7 to the same style. Since the magazine is published mainly in electronic form, color drawings would look great, especially where it is necessary to mark different groups (sex, age, location of capture).

Author Response

Dear Reviewer,

Thank you for spending your valuable time to reviewing our manuscript. 

Here is the attached point-by-point response to your comments.

Best regards,

Abdou.

Reviewer 3 Report

This is a straightforward paper, well written and providing data from a poorly studied region.

Most of the problems I detect are with formatting.

Many scientific names are not italicized, even in the bibliography

194-196. These lines repeat the data in Table 1.

409-412. The tense wrong: The prevalence of P. merus in Benin (45%) was higher than THAT in Ghana (21%) during the study. The prevalence of this species in Benin and Ghana WAS lower than THAT recorded FROM S. aurita in Bizerte (84%), Kelibia (84.44%), Mahdia (48.05%) and Zarzis 411 (86.84%) off the coast of Tunisia [21] but higher than THAT recorded FROM S. aurita from Ga …

467. Why not describe the new Mazocraeoides species? This would increase greatly the value of the study. Ecological studies on unidentified species are of reduced value.

Table 2. The sites of Infection V and G are not listed in the legend.

Figures 4 and 5. The keys to the different histogram columns are much too small and the colours are not discernible. I suggest these be greatly enlarged and coloured more distinctly.

Figure 6. The label on the Y axis is the name of the parasite, not the parameter in question.

The references are formatted strangely. The words in titles of the articles have mostly capitalized words. Usually in English, capitalization is only used on at the start of the sentence and on nouns. Throughout the bibliography the specific names are capitalized – this is incorrect, e.g., Sardinella Maderensis should be Sardinella maderensis.

Reference 23 has no Journal name. It is Atlântica, Rio Grande.

Author Response

Dear Reviewer,

Thank you for accepting spend your valuable time to review our manuscript.

Kindly find attached the point-by-point response to your comments.

With best regards,

Abdou.
